# AMORTIZED CONDITIONAL NORMALIZED MAXIMUM LIKELIHOOD

## ABSTRACT

While deep neural networks provide good performance for a range of challenging tasks, calibration and uncertainty estimation remain major challenges. In this paper, we propose the amortized conditional normalized maximum likelihood (ACNML) method as a scalable general-purpose approach for uncertainty estimation, calibration, and out-of-distribution robustness with deep networks. Our algorithm builds on the conditional normalized maximum likelihood (CNML) coding scheme, which has minimax optimal properties according to the minimum description length principle, but is computationally intractable to evaluate exactly for all but the simplest of model classes. We propose to use approximate Bayesian inference technqiues to produce a tractable approximation to the CNML distribution. Our approach can be combined with any approximate inference algorithm that provides tractable posterior densities over model parameters. We demonstrate that ACNML compares favorably to a number of prior techniques for uncertainty estimation in terms of calibration on out-of-distribution inputs.

## 1 INTRODUCTION

Current machine learning methods provide unprecedented accuracy across a range of domains, from computer vision to natural language processing. However, in many high-stakes applications, such as medical diagnosis or autonomous driving, rare mistakes can be extremely costly, and thus effective deployment of learned models requires not only high expected accuracy, but also a way to measure the certainty in a model's predictions in order to assess risk and allow the model to abstain from making decisions when there is low confidence in the prediction. While deep networks offer excellent prediction accuracy, they generally do not provide the means to accurately quantify their uncertainty. This is especially true on out-of-distribution inputs, where deep networks tend to make overconfident incorrect predictions (Ovadia et al., 2019). In this paper, we tackle the problem of obtaining reliable uncertainty estimates under distribution shift.

Most prior work approaches the problem of uncertainty estimation from the standpoint of Bayesian inference. By treating parameters as random variables with some prior distribution, Bayesian inference can compute posterior distributions that capture a notion of *epistemic* uncertainty and allow us to quantitatively reason about uncertainty in model predictions. However, computing accurate posterior distributions becomes intractable as we use very complex models like deep neural nets, and current approaches require highly approximate inference methods that fall short of the promise of full Bayesian modeling in practice.

Bayesian methods also have a deep connection with the minimum description length (MDL) principle, a formalization of Occam's razor that recasts learning as performing efficient lossless data compression and has been widely used as a motivation for model selection techniques. Codes corresponding to maximum-a-posteriori estimators and Bayes marginal likelihoods have been commonly used within the MDL framework. However, other coding schemes have been proposed in MDL centered around achieving different notions of minimax optimality. Interpreting coding schemes as predictive distributions, such methods can directly inspire prediction strategies that give conservative predictions and do not suffer from excessive overconfidence due to their minimax formulation.

One such predictive distribution is the *conditional normalized maximum likelihood* (CNML) (Grünwald, 2007; Rissanen and Roos, 2007; Roos et al., 2008) model, also known as sequential NML or predictive NML (Fogel and Feder, 2018b). To make a prediction on a new input, CNML considers

every possible label and tries to find the model that best explains that label for the query point together with the training set. It then uses that corresponding model to assign probabilities for each input and normalizes to obtain a valid probability distribution. Intuitively, instead of relying on a learned model to extrapolate from the training set to the new (potentially out-of-distribution) input, CNML can obtain more reasonable predictive distributions by asking "given the training data, which labels would make sense for this input?"

While CNML provides compelling minimax regret guarantees, practical instantiations have been exceptionally difficult, because computing predictions for a test point requires retraining the model on the test point *concatenated with the entire training set*. With large models like deep neural networks, this can potentially require hours of training for every prediction.

In this paper, we proposed *amortized CNML* (ACNML), a tractable and practical algorithm for approximating CNML utilizing approximate Bayesian inference. ACNML avoids the need to optimize over large datasets during inference by using an approximate posterior in place of the training set. We demonstrate that our proposed approach is substantially more feasible and computationally efficient than prior techniques for using CNML predictions with deep neural networks and compares favorably to a number of prior techniques for uncertainty estimation on out-of-distribution inputs.

## 2 MINIMUM DESCRIPTION LENGTH: BACKGROUND AND PRELIMINARIES

ACNML is motivated from the minimum description length (MDL) principle, which can be used to derive a connection between optimal codes and prediction. We begin with a review of the MDL principle and discuss the challenges in implementing minimax codes that motivate our method. For more comprehensive treatments of MDL, we refer the readers to (Grünwald, 2007; Rissanen, 1989).

**Minimum description length.** The MDL principle states that any regularities in a dataset can be exploited to compress it, and hence learning is reformulated as losslessly transmitting the data with the fewest number of bits (Rissanen, 1989; Grünwald, 2007). Simplicity is thus formalized as the length of the resulting description. MDL was originally formulated in a generative setting where the goal is to code arbitrary data, and we will present a brief overview in this setting. We can translate the results to a supervised learning setting, which corresponds to transmitting the labels after assuming either a fixed coding scheme for the inputs or that the inputs are known beforehand. While MDL is typically described in terms of code lengths, in general, we can associate codes with probability distributions, with the code length of an object corresponding to the negative log-likelihood under that probability distribution (Cover and Thomas, 2006).

**Normalized Maximum Likelihood.** Let $\hat{\theta}(x_{1:n})$ denote the maximum likelihood estimator for a sequence of data $x_{1:n}$ over all $\theta \in \Theta$. For any $x_{1:n} \in \mathcal{X}^n$ and distribution $q$ over $\mathcal{X}^n$, we can define a regret relative to the model class $\Theta$ as

$$R(q, \Theta, x_{1:n}) \stackrel{\text{def}}{=} \log p_{\hat{\theta}(x_{1:n})}(x_{1:n}) - \log q(x_{1:n}). \tag{1}$$

This regret corresponds to the excess number of bits $q$ uses to encode $x_{1:n}$ compared to the best distribution in $\Theta$, denoted $\hat{\theta}(x_{1:n})$. We can then define the *normalized maximum likelihood distribution* (NML) with respect to $\Theta$ as

$$p^{\text{NML}}(x_{1:n}) = \frac{p_{\hat{\theta}(x_{1:n})}(x_{1:n})}{\sum_{\tilde{x}_{1:n} \in \mathcal{X}^n} p_{\hat{\theta}(\tilde{x}_{1:n})}(\tilde{x}_{1:n})} \tag{2}$$

when the denominator is finite. The NML distribution can be shown to achieve minimax regret (Shtarkov, 1987; Rissanen, 1996)

$$p^{\text{NML}} = \underset{q}{\text{argmin}} \max_{x_{1:n} \in \mathcal{X}^n} R(q, \Theta, x_{1:n}). \tag{3}$$

This corresponds, in a sense, to an optimal coding scheme for sequences of known fixed length.

**Conditional NML.** Instead of making predictions across entire sequences at once, we can adapt NML to the setting where we make predictions about the next data point based on the previously seen data, resulting in *conditional NML* (CNML) (Rissanen and Roos, 2007; Grünwald, 2007; Fogel and Feder, 2018a). While several variations on CNML exist, we consider the following:

$$p^{\text{CNML}}(x_n | x_{1:n-1}) \propto p_{\hat{\theta}(x_{1:n})}(x_n). \tag{4}$$

For any fixed sequence $x_{1:n-1}$, $p^{\text{CNML}}$ solves the minimax regret problem

$$p^{\text{CNML}} = \underset{q}{\arg\min} \max_{x_n} \log p_{\hat{\theta}(x_{1:n})}(x_n) - \log q(x_n), \tag{5}$$

where the inner maximization is only over the last data point $x_n$.

We can extend this approach to the supervised classification setting, where our models represent conditional distributions $p_\theta(y|x)$. The CNML distribution, given a sequence of already seen datapoints $(x_{1:n-1}, y_{1:n-1})$ and the next input $x_n$, then takes the form

$$p^{\text{CNML}}(y_n|x_n; x_{1:n-1}, y_{1:n-1}) \propto p_{\hat{\theta}(y_{1:n}|x_{1:n})}(y_n|x_n), \tag{6}$$

and solves the minimax problem

$$p^{\text{CNML}} = \underset{q}{\arg\min} \max_{y_n} \log p_{\hat{\theta}(y_{1:n}|x_{1:n})}(y_n|x_n) - \log q(y_n). \tag{7}$$

We see that this conditional distribution is amenable to our usual inductive learning procedure, where $(x_{1:n-1}, y_{1:n-1})$ is our training set, and we want to output a predictive distribution over labels $y_n$ for a new test input $x_n$.

**CNML provides conservative predictions.** For each query point, CNML considers each potential label and finds the model that would be most consistent with that label and with the training set. If that model assigns high probability to the label, then minimizing the worst-case regret forces CNML to assign relatively high probability to it. In particular, compared to simply letting a model trained only on the training set extrapolate blindly, we expect CNML to give more conservative predictions on out-of-distribution inputs, since it explicitly considers what would have happened if the new data point had been included in the training dataset with each particular label.

We use a 2D logistic regression example to illustrate CNML's conservative predictions, showing a heatmap of CNML probabilities in Figure 1. CNML provides uniform predictions on most of the input space away from the training samples. In Figure 2, we illustrate how CNML arrives at these predictions, showing the predictions for the parameters $\hat{\theta}_0$ and $\hat{\theta}_1$, corresponding to labeling the test point (shown in pink in Figure 2, left) with either the label 0 or 1.

However, CNML may be too conservative when the model class $\Theta$ is very expressive. Naïvely applying CNML with large model classes can result in the per-label models fitting their labels for the query point arbitrarily well, such that CNML gives unhelpful uniform predictions even on inputs we

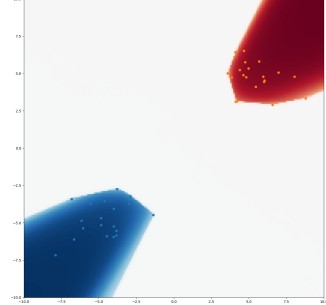

**Figure 1:** CNML probabilities with a logistic regression model. Note that CNML provides uniform predictions (indicated by the white color) on most of the input space away from the training set (shown in blue and orange dots).

would hope to reasonably extrapolate on. We see this in the 2D logistic regression example in Figure 1. Thus, the model class $\Theta$ would need to be restricted in some form, for example by only considering only parameters within a certain distance from the training set solution as a hard constraint.

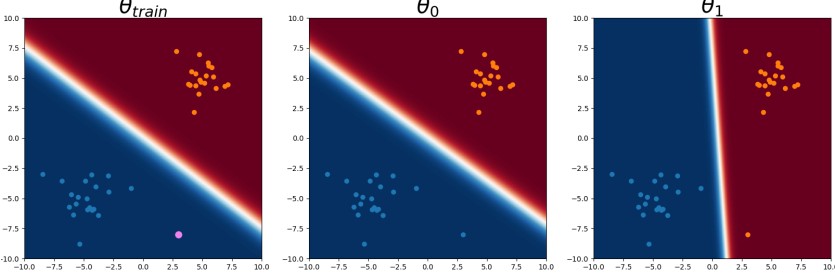

**Figure 2:** Given the labeled training set (blue and orange dots), we want to predict the label at the query input (shown in pink in the left image), which the training set MLE $\hat{\theta}_{\text{train}}$ confidently classifies as the blue class. However, CNML assigns a near-uniform prediction on the query point, as it computes new MLEs $\hat{\theta}_0$ and $\hat{\theta}_1$ (center and right images) by assigning different labels to the query point, and finds both labels are consistent with the training data.

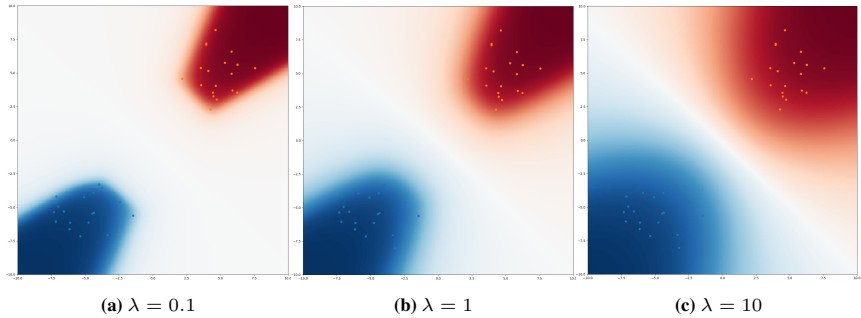

(a) $\lambda = 0.1$        (b) $\lambda = 1$        (c) $\lambda = 10$

**Figure 3:** CNMAP probability heatmaps with different levels of L2 regularization $\lambda \|w\|_2^2$. We see predictions are less conservative as regularization increases.

Another approach for controlling the expressivity of the model class is to generalize CNML to use *regularized* estimators instead of maximum likelihood, resulting in normalized maximum a posteriori (NMAP) (Kakade et al., 2006) codes. Instead of using maximum likelihood parameters, NMAP selects $\hat{\theta}$s to be the parameter that maximizes both data likelihood and a regularization term, or prior, over parameters, and we can define slightly altered notions of regret using these MAP estimators in all the previous equations to get a *conditional normalized maximum a posteriori* distribution instead. See Appendix D for completeness.

Going back to the logistic regression example, we plot heatmaps of CNMAP predictions in Figure 3, adding different amounts of L2 regularization to the logistic regression weights. As we add more regularization, the model class becomes effectively less expressive, and the CNMAP predictions become less conservative.

**Computational Costs of CNML.** A major practical issue with actually utilizing CNML or CNMAP with neural networks is the prohibitive computational costs of computing the maximum likelihood estimators for each new input and label combination. To evaluate the distribution on a new test point, one must solve a nonconvex optimization problem for each possible label, with each problem involving the entire training dataset along with the new test point. This direct evaluation of CNML therefore becomes computationally infeasible with large datasets and high-capacity models, and further requires that the model carry around the entire training set even when it is deployed. In settings where critical decisions must be made in real time, even running a single epoch of additional training would be infeasible. For this reason, NML-based methods have not gained much traction as a practical tool for improving the predictive performance of high-capacity models.

## 3    AMORTIZED CNML

In this section, we derive our method, amortized conditional normalized maximum likelihood (ACNML). ACNML provides a tractable approximation for CNML and CNMAP via approximate Bayesian inference. Instead of directly computing maximum likelihood parameters over the query point and training set, our method uses an approximate posterior distribution over parameters to capture the necessary information about the training set, and thus reduces the maximization to only the single new point. The computational cost at test-time therefore does not increase with training set size. We specialize our notation to the supervised learning setting, where our aim is to obtain a predictive distribution $p(y_n|x_n)$ after observing a training set $(x_{1:n-1}, y_{1:n-1})$ and a test input $x_n$.

### 3.1    ALGORITHM DERIVATION

**Incorporating an exact posterior into CNML.** Given a prior distribution $p(\theta)$, the Bayesian posterior likelihood conditioned on the training data is given by

$$p(\theta|x_{1:n-1}, y_{1:n-1}) \propto p(\theta)p_\theta(y_{1:n-1}|x_{1:n-1}). \tag{8}$$

We can write the MAP estimators in the CNMAP distribution for a fixed query input $x_n$ as

$$\hat{\theta}_y = \underset{\theta \in \Theta}{\arg\max} \log p_\theta(y|x_n) + \underbrace{\log p_\theta(y_{1:n-1}|x_{1:n-1}) + \log p(\theta)}_{\log p(\theta|x_{1:n-1}, y_{1:n-1})} \tag{9}$$

---

**Algorithm 1** Amortized CNML (ACNML)

---

**Input**: Model class $\Theta$, Training Data $(x_{1:n-1}, y_{1:n-1})$, Test Point: $x_n$, Classes $(1, \ldots, k)$
**Output**: Predictive distribution $p(y|x_n)$
**Training**: Run approximate inference algorithm on training data $(x_{1:n-1}, y_{1:n-1})$ to get posterior density $q(\theta)$
**for all** possible labels $i \in (1, \ldots, k)$ **do**
    Compute $\hat{\theta}_i = \text{argmax}_\theta \log p_\theta(i|x_n) + \log q(\theta)$
**end for**
Return $p(y|x_n) = \frac{p_{\hat{\theta}_y}(y|x_n)}{\sum_{i=1}^{k} p_{\hat{\theta}_i}(i|x_n)}$

---

We can thus replace the training data log-likelihood $p_\theta(y_{1:n-1}|x_{1:n-1})$ with the Bayesian posterior density $\log p(\theta|x_{1:n-1}, y_{1:n-1})$ when computing $\hat{\theta}_y$. We can also recover CNML as a special case of CNMAP by using a uniform prior, but as discussed previously, CNML with highly expressive model classes can lead to overly conservative predictions, so we will opt to use non-uniform priors that help control model complexity instead. For example, with deep neural networks, we may elect to use a zero-mean Gaussian prior $p(\theta)$ on the network weights, corresponding to L2 regularization.

**ACNML with an approximate posterior.** Of course, the exact Bayesian likelihood is no easier to compute than the original training log likelihood. However, we can derive a tractable approximation by replacing the exact posterior $p(\theta|x_{1:n-1}, y_{1:n-1})$ with an approximate posterior $q(\theta)$ instead. We can obtain an approximate posteriors via standard approximate Bayesian techniques such as variational inference or Laplace approximations. We focus on Gaussian posterior approximations for computational efficiency, and discuss in Section 3.2 why this class of distributions provides a reasonable approximation.

For practical purposes, we expect the approximate posterior log-likelihood to ensure the optimal $\hat{\theta}_y$ selected for each label retains good performance on the training set. By replacing the likelihood over the training data with the probability under an approximate posterior, it becomes unnecessary to retain the training data at test time, only the parameters of the approximate distribution. Optimization also becomes much simpler, as it no longer requires stochastic gradients, and the Gaussian posterior log density $\log q(\theta)$ can serve as a strong convex regularizer.

**ACNML algorithm summary** A summary of the ACNML algorithm is presented in Algorithm 1. The training process for obtaining $q(\theta)$ only needs to be performed once on the training set, whereas the inference step is performed for each test point. However, this inference step only requires optimizing the model parameters on a single data point, with the regularizer provided by $\log q(\theta)$.

### 3.2 ANALYSIS OF GAUSSIAN APPROXIMATIONS IN ACNML

In this section, we argue that using a Gaussian approximate posteriors in ACNML, which correspond to second-order approximations to the training set log-likelihood, suffice for accurately computing the CNML distributions when the training set is large. The intuition is that for large training sets, the combined likelihoods of all the training points dominate over the single new test point, so the perturbed MLEs $\hat{\theta}_y$ remains close to the original training set MLE $\hat{\theta}$, letting us rely on local approximations to the training loss.

Under some simplifying assumptions, we can formalize this argument using the concept of *influence functions*, which measure how maximum likelihood parameters (and more general $M$-estimators) for a dataset would change if the dataset were perturbed by reweighting inputs an infinitesimal amount.

We recall that maximum likelihood estimators for a dataset with $n$ datapoints $(x_{1:n}, y_{1:n})$ is given by

$$\hat{\theta} = \underset{\theta}{\text{argmax}} \frac{1}{n} \sum_{i=1}^{n} \log p_\theta(y_i|x_i). \tag{10}$$

Influence functions analyze how $\hat{\theta}$ relates to the MLE of a perturbed dataset

$$\hat{\theta}_{x,y,\epsilon} = \underset{\theta}{\text{argmax}} \left( \epsilon \log p_\theta(y|x) + \frac{1}{n} \sum_{i=1}^{n} \log p_\theta(y_i|x_i) \right), \tag{11}$$

where $\hat{\theta}_{x,y,\epsilon}$ is the new MLE if we perturb the training set by adding a datapoint $(x, y)$ with a weight $\epsilon$. A classical result (Cook and Weisberg, 1982) shows that $\hat{\theta}_{x,y,\epsilon}$ is differentiable (under appropriate regularity conditions) with respect to $\epsilon$ with derivative given by the influence function

$$\frac{d\hat{\theta}_{x,y,\epsilon}}{d\epsilon}|_{\epsilon=0} = -H_{\hat{\theta}}^{-1} \nabla_\theta \log p_{\hat{\theta}}(y|x), \tag{12}$$

where $\hat{\theta}$ is the MLE for the original dataset and $H_{\hat{\theta}}$ the Hessian of the mean training set log-likelihood evaluated at $\hat{\theta}$. CNML computes the MLE after adding datapoint $(x, y)$ with equal weight to points in the training set, which is precisely given by $\hat{\theta}_{x,y,\epsilon}$ evaluated at $\epsilon = 1/n$. Thus, for sufficiently large $n$, a first order Taylor expansion around $\hat{\theta}$ should be accurate and the new parameter can be estimated by

$$\tilde{\theta}_{x,y} = \hat{\theta} - \frac{1}{n} H_{\hat{\theta}}^{-1} \nabla_\theta \log p_{\hat{\theta}}(y|x), \tag{13}$$

which is equivalent to solving

$$\tilde{\theta}_{x,y} = \underset{\theta}{\mathrm{argmax}} \frac{1}{n}(\theta - \hat{\theta})^T \nabla_\theta \log p_{\hat{\theta}}(y|x) + \frac{1}{2}(\theta - \hat{\theta})^T H_{\hat{\theta}}(\theta - \hat{\theta}). \tag{14}$$

This suggests that, with large training datasets, the perturbed MLE parameters $\hat{\theta}_y$ in Equation 9 can be approximated accurately using a quadratic approximation to the training log-likelihood, corresponding to a Gaussian posterior obtained via a Laplace approximation. We can explicitly quantify the accuracy of this approximation in the theorem below, which is based on Theorem 1 from Giordano et al. (2019), with full details and proof in Appendix E.

**Theorem 3.1.** *(Adapted from Giordano et al. (2019)) Consider a training set with $n$ datapoints and an additional datapoint $(x, y)$. Assume assumptions 1-5 hold with constants $C_{op}, C_{IJ}, \Delta_\delta$ as defined in Appendix E. Let $\hat{\theta}_{x,y}$ denote the exact MLE if we had appended $(x, y)$ to the training set, and $\tilde{\theta}_{x,y}$ the parameter obtained via the approximation in Equation 13.*

*Let*

$$\delta = \frac{1}{n+2} \max\{\sup_{\theta \in \Theta} \|\nabla_\theta \log p_\theta(y|x)\|_1, \sup_{\theta \in \Theta} \|\nabla_\theta^2 \log p_\theta(y|x)\|_1\}. \tag{15}$$

*If $\delta \leq \Delta_\delta$, then*

$$\left\| \hat{\theta}_{x,y} - \tilde{\theta}_{x,y} \right\|_2 \leq 2C_{op}^2 C_{IJ}\delta^2, \tag{16}$$

Given a bound on how accurately we estimate the new parameters for CNML, we can also explicitly quantify the accuracy of the resulting normalized distributions, with proof in Appendix E.

**Proposition 3.2.** *Suppose $y \in \mathcal{Y}$ with $|\mathcal{Y}| = k$ (classification with $k$ classes). Let $\theta_{x,y}$ be the exact MLE after appending the datapoint $(x, y)$ to the training set, and let $\tilde{\theta}_{x,y}$ be an approximate MLE with $\left\| \hat{\theta}_{x,y} - \tilde{\theta}_{x,y} \right\| \leq \delta$ for each $y$. Further suppose $\log p_\theta(y|x)$ is $L$-Lipschitz in $\theta$.*

*Denote the exact CNML distribution for the fixed input $x$ to be $p_{CNML}(y) \propto p_{\hat{\theta}_{x,y}}(y|x)$ and an approximate CNML distribution $p_{ACNML}(y) \propto p_{\tilde{\theta}_{x,y}}(y|x)$. We then have*

$$\sup_y |\log p_{CNML}(y) - \log p_{ACNML}(y)| \leq 2L\delta. \tag{17}$$

Theorem 3.1 and Proposition 3.2 together suggest that the approximation produced by ACNML will be increasingly close to the exact CNML distribution as the training set size $n$ grows. However, this formal theoretical result only holds for sufficiently large datasets and under strong simplifying assumptions including smoothness and strong convexity of the training loss, so **does not necessarily hold in practical settings with deep neural networks**.

In the context of interpreting how different data points influence the predictions of neural networks, Koh and Liang showed that influence function approximations were able to provide useful predictions for estimating leave-one-out retraining with deep convolutional neural networks. This closely resembles the conditions we encounter when computing parameters for each label of the query point with ACNML, with the key difference being that ACNML *adds* a datapoint while leave-one-out retraining *removes* one. This suggests second-order approximations to the training loss, corresponding to Gaussian approximations in ACNML, may suffice to yield useful predictions about how parameters change when the query point is added, despite lacking formal guarantees with deep neural networks.

## 4  RELATED WORK

Minimum description length has been used to motivate neural net methods dating back to Hinton and van Camp (1993), who treat description length as a regularizer to mitigate overfitting. The idea of preferring flat minima (Hochreiter and Schmidhuber, 1997) also has its origins in the MDL framework, as it allows a coarser discretization of the weights (and thus fewer bits needed).

Bayesian methods typically serve as the starting point for uncertainty estimation in deep networks, and a commonly used approach is to use simple tractable distributions to approximate the true posterior (Hoffman et al., 2013; Blundell et al., 2015; Ritter et al., 2018). Recent work (Maddox et al., 2019; Dusenberry et al., 2020) has shown fairly simple posterior approximations are able to achieve well-calibrated predictions with marginalization. Our method builds on top of these approximate posterior methods, but in contrast to the Bayesian methods, where the posterior is typically used to efficiently sample models for Bayesian model averaging, our method uses the posterior density to enable efficient optimization for computing the CNML, without needing to retain the training data.

Ovadia et al. (2019) evaluate various proposed methods for uncertainty estimates in deep learning under different types of distribution shift. They found that good calibration on in-distribution points did not necessarily indicate good calibration under distribution shift, and that methods relying on marginalizing predictions over multiple models (Lakshminarayanan et al., 2016; Srivastava et al., 2014) gave better uncertainty estimates under distribution shift than other techniques. We show that our method ACNML maintains much better calibration under distribution shift than prior methods.

Perhaps most closely related to our work, Fogel and Feder (2018b) advocate for the use of the CNML distribution in the context of supervised learning (under the name predictive NML), citing its minimax properties. Bibas et al. (2019b) estimate the CNML distribution with deep networks by finetuning the last layers of the network on every test input and label combination appended to the training set. Since this finetuning procedure trains for several epochs, it is very computationally intensive at test-time and requires continued access to the entire training set when evaluating. In contrast, our method amortizes this procedure by condensing the information in the training data into a distribution over parameters, allowing for much faster test-time inference without needing the training data.

In the analysis for our approximation, we use influence functions (Cook and Weisberg, 1982), which have been studied as asymptotic approximations to how $M$-estimators change when perturbing a dataset. In deep learning, Koh and Liang advocated for using influence functions to interpret neural nets, generate adversarial examples, and diagnose errors in datasets. We use a theorem from Giordano et al. (2019), which broadened the necessary assumptions for these infinitisemal approximations to be accurate and provides explicit guarantees for fixed datasets rather than asymptotic results.

## 5  EXPERIMENTS

To instantiate ACNML, we must select a method for obtaining the approximate posterior. In principle, any technique for computing a tractable posterior over parameters can be used, and we demonstrate this flexibility by implementing ACNML on top of Stochastic Weight Averaging - Gaussian (SWAG) (Maddox et al., 2019), KFAC-Laplace (Ritter et al., 2018), and Bayes-by-backprop (Blundell et al., 2015). SWAG computes a posterior by fitting a Gaussian distribution to the trajectory of SGD iterates. For simplicity and computational efficiency, we instantiate ACNML with the SWAG-D variant, which uses a Gaussian posterior with only a diagonal covariance. KFAC-Laplace uses a Gaussian posterior approximation with the MAP solution as the mean and the inverse Hessian of the negative log likelihood as covariance, approximating the Hessian using KFAC (Martens and Grosse, 2015) to allow for tractable inversion and storage. Bayes-by-backprop (Blundell et al., 2015) uses the reparameterization trick to learn a diagonal Gaussian posterior via the variational lower bound.

For each model, we report results across 3 seeds. We compare negative log likelihood (NLL), accuracy, and expected calibration error (ECE) (Naeini et al., 2015) as well as showing reliability diagrams (Guo et al., 2017) to further assess calibration. For reliability diagrams, we sort data points by confidence and divide them into twenty equal sized buckets, plotting the mean accuracy against the mean confidence for each bucket. This allows to see qualitatively see how well the confidence of the prediction relates to the actual accuracy, as well as showing how the confidences are distributed for each method.

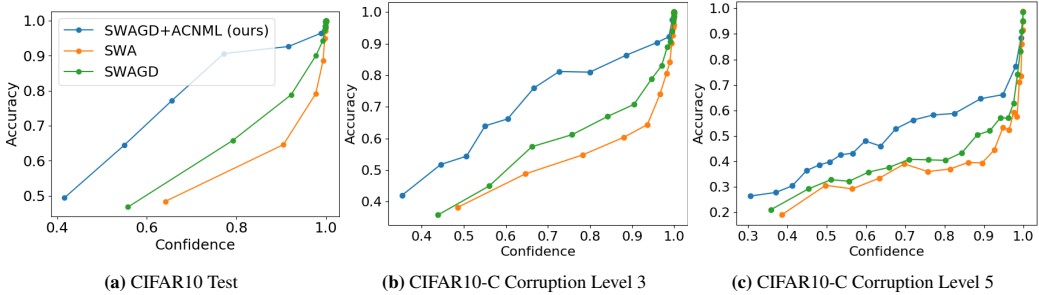

(a) CIFAR10 Test          (b) CIFAR10-C Corruption Level 3          (c) CIFAR10-C Corruption Level 5

**Figure 4:** Reliability diagrams plotting confidence vs. accuracy for VGG16 on in-distribution and out-of-distribution data. ACNML provides more conservative predictions than other methods, resulting in better calibration on out-of-distribution inputs. For the OOD task, we show results for the Gaussian blur corruptions at levels 3 and 5, with level 5 corresponding to a higher amount of corruption. Each point shows the mean confidence and mean accuracy within a bucket, so the spread of points along the $x$-axis shows that ACNML makes more low confidence predictions than other methods.

**MNIST.** We start with a simple illustrative task based on the MNIST dataset, where we construct out-of-distribution inputs by randomly rotating the images in the MNIST test set. Here, ACNML is implemented on top of Bayes-by-backprop (Blundell et al.,

| | MNIST | | | Rotated MNIST (OOD) | | |
|---|---|---|---|---|---|---|
| | NLL | Accuracy | ECE | NLL | Accuracy | ECE |
| ACNML (ours) | 0.1818 | 97.28\$ | 0.1013 | **2.766** | 37.34 | **0.1540** |
| MAP | 0.0864 | 97.28 | **0.0047** | 3.994 | 37.2 | 0.4371 |
| Marginal | 0.1069 | 97.22 | 0.0313 | 3.017 | 37.63 | 0.2928 |
| naive CNML | **0.0774** | **98.05** | 0.0231 | 3.100 | 37.33 | 0.2497 |

**Table 1:** Comparative results for ACNML on MNIST using a posterior obtained via Bayes by Backprop.

2015), and we compare to the MAP estimate and the marginal over models obtained from the same Bayes-by-backprop posterior. The results in Table 1 show that all methods perform well on the in-distribution MNIST test set, though ACNML likelihoods are somewhat worse due to the more conservative CNML distribution. On OOD rotated digits, we see that ACNML exhibits substantial improvements in calibration as measured by the ECE metric, as well as slightly better NLL value. In general, this agrees with what we expect from ACNML: the predictions are more conservative across the board, which does not necessarily improve results in-distribution, particularly for easy domains like MNIST, but offer considerable improvements in calibration for out-of-distribution inputs where errors are prevalent. We additionally compared to a much more computationally expensive instantiation of CNML used by Bibas et al. (2019a) (denoted naive CNML in Table 1), which directly finetunes for several epochs using the training set to obtain the optimal parameters for each query point and label, rather than using the approximate posterior like ACNML does. This direct instantiation of CNML performs the best in terms of accuracy and NLL on the in-distribution test set, while also improving over the MAP solution in terms of NLL and calibration on the OOD inputs. However, we find that ACNML is overall more conservative when using this particular posterior approximation, resulting in better NLL and calibration on the OOD inputs (see Appendix C for more detailed comparisons between ACNML and naive CNML).

**CIFAR and Corruptions** We evaluate all methods using the VGG16 (Simonyan and Zisserman, 2014) network architecture. Focusing on the most direct comparisons, we compare against the MAP solution for the given posterior, which is equivalent to Stochastic Weight Averaging (SWA) (Izmailov et al., 2018), and Bayes model averaging with SWAGD and KFAC-Laplace, which provide an apples-to-apples comparison to the two versions of our method that directly utilize the posteriors from these prior approaches. We use CIFAR10 (Krizhevsky, 2012) for training and in-distribution testing. Following (Ovadia et al., 2019), we evaluate predictive uncertainty in out-of-distribution settings using the CIFAR10-Corrupted (Hendrycks and Dietterich, 2019) datasets, which apply different severities of 15 common corruptions to the test set images. With this, we can assess performance over a wide range of distribution shifts, as well as how performance degrades as shifts become more extreme. We include additional comparisons across other methods and architectures in Appendix B.

Examining the reliability diagrams in Figure 4, we see that ACNML provides more conservative (less confident) predictions than other methods, to the point of being *underconfident* on the in-distribution CIFAR10 test set, while other methods tend toward being overconfident. On out-of-distribution datasets, where accuracy degrades, we see that ACNML's conservative predictions lead to many better calibrated low-confidence predictions, while other methods drastically overestimate confidence.

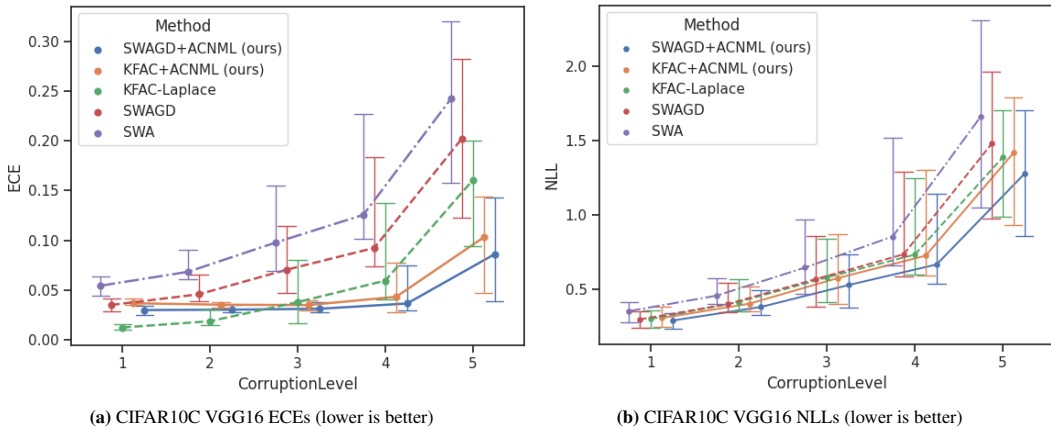

**(a)** CIFAR10C VGG16 ECEs (lower is better)  **(b)** CIFAR10C VGG16 NLLs (lower is better)

**Figure 5:** ACNML compared against their Bayesian counterparts and the deterministic MAP baseline on out-of-distribution CIFAR10-Corrupted datsets. We plot medians and 95% confidence intervals across all corruptions. We see that ACNML methods (solid lines) achieve much lower ECE at higher corruption values, and ACNML with SWAGD also achieves better NLL than other methods.

All methods perform similarly in terms of accuracy in all domains, and we find that ACNML's more conservative estimates perform competitively with Bayesian methods in NLL and calibration on in-distribution datasets, with all evaluated methods performing reasonably well in-distribution (see Table 3 in Appendix B). However, differences in calibration are much more pronounced for the OOD results in Figure 5. We see that as the corruption strength increases, ACNML variants provide much better calibration while performing similarly to or slightly better than other methods in terms of NLL.

**Timing Comparison vs. standard CNML**: In Table 2, we examine the computational costs of our method. We compare against a naïve implementation of CNML that fine-tunes for $N$ epochs on each test point and label, similarly to the method proposed

|  | MNIST MLP | VGG16 | WRN28x10 |
|---|---|---|---|
| ACNML (ours) | 0.08s | 0.37s | 1.1s |
| naïve CNML (per epoch) | 13.83s | 102.0s | 359.1s |
| feedforward inference | 0.0001s | 0.0013s | 0.004s |

**Table 2:** Inference time per input (in seconds).

by Bibas et al. (2019b). In total, predicting a single input with $k$ possible labels involves running $kN$ epochs of training. While ACNML is over two orders of magnitude faster than naïve CNML even with just a single epoch of training (our experiments with naive CNML on MNIST used 5 epochs), it is still slower than standard inference. The computational requirements of our method scale linearly with the number of classes, but are constant with respect to dataset size. It is also not easily amenable to data batching, as new copies of the model parameters are needed for each data point. Timing experiments are run using a single NVIDIA 1080Ti, using MNIST for the MNIST MLP timing reselts and using CIFAR10 for VGG16 and WideResNet28x10, with no parallelization over data points.

## 6  DISCUSSION

In this paper, we present amortized CNML (ACNML) as an alternative to Bayesian marginalization for obtaining uncertainty estimates and calibrated predictions with high-capacity models, such as deep neural networks. The CNML distribution is a theoretically well-motivated strategy derived from the MDL principle with strong minimax optimality properties, but actually evaluating this distribution is computationally daunting. ACNML utilizes approximate Bayesian posteriors to tractably approximate it, and can be instantiated on top of a wide range of approximate Bayesian methods. We view ACNML as a step towards practical uncertainty aware predictions that would be essential for real-world decision making. Future work could further improve on our proposed method, for example by combining ACNML with more complex and expressive posterior approximations. In particular, training losses are highly non-convex and have many local minima, so incorporating local approximations around *multiple* diverse minima could allow for even more reliable uncertainty estimation. More broadly, tractable algorithms inspired by ACNML could in the future provide for substantial improvement in our ability to produce accurate and reliable confidence estimates on out-of-distribution inputs, improving the reliability and safety of learning-enabled systems.

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

## A  EXPERIMENTAL DETAILS

For obtaining approximate posteriors with SWAG and KFAC-Laplace, we follow the exact training procedures given in Maddox et al. (2019). We then implement ACNML on top of the diagonal SWAG posterior and the KFAC-Laplace posterior.

The variance of the SWAG posterior depends in a complex way on the learning rate and gradient covariances. To account for this, we introduce an additional temperature hyperparameter $\alpha$ and solve for the ACNML approximation using

$$\theta^* = \underset{\theta \in \Theta}{\operatorname{argmax}} \log p_\theta(y_n|x_n) + \frac{1}{\alpha} \log q(\theta). \tag{18}$$

To calibrate $\alpha$, we can calculate the CNML distribution using a validation set, by training on the entire training set and the validation point, and then selecting $\alpha$ such that our ACNML procedure produces similar likelihoods. We can also treat $\alpha$ as a tunable hyperparameter and select it using a validation set, similarly to how temperature scaling (Guo et al., 2017) is used to achieve better calibration for prediction, or how the relative weighting of priors and likelihoods are used in generalized Bayesian inference (Vovk, 1990) or safe Bayesian inference (Grünwald et al., 2017) as a way to deal with model misspecification. For our experiments using the SWAGD posterior, we heuristically tune $\alpha$ to be as large as possible without degrading the accuracy compared to the MAP solution. Note, however, that this procedure is specific to the particular way in which SWAG estimates the parameter distribution, and any posterior inference procedure that explicitly approximates the posterior likelihood (e.g., Blundell et al. (2015)) would not require this step. To select $\alpha$ for each model class, we swept over values $[0.25, 0.5, 1, 1.5, 2]$ and selected the highest value such that accuracy and NLL on the validation set did not degrade significantly compared to SWA. For VGG16, we use $\alpha = 0.5$ and for WideResNet28x10, we used $\alpha = 1.5$.

With our posterior $q(\theta)$ being a Gaussian with covariance $\Sigma$, we approximately compute the MAP solution for each label $y$ as per Algorithm 1 by initializing $\theta_0$ to be the posterior mean and iterating

$$\theta_{t+1} = \theta_t + \epsilon_t \Sigma(\alpha \nabla \log p_{\theta_t}(y|x_n) + \nabla \log q(\theta_t)), \tag{19}$$

using the covariance as a preconditioner. For our experiments, we run 5 steps of gradient ascent on this objective, with a constant step size $\epsilon = 0.5$. We empirically find that 5 steps was often enough to find an approximate stationary point with the SWAG-D posterior, and 10 steps for the KFAC-Laplace posterior.

For the reliability diagrams in Figure 4, we again follow the procedure used by Maddox et al. (2019). We first divide the points into twenty bins uniformly based on confidence (each bin has the same number of points), then plot the mean accuracy vs mean confidence within each bin. This differs from the reliability diagrams used by Guo et al. (2017), where they divide the range of confidence values into bins uniformly, resulting in unevenly filled bins.

For our expected calibration error (ECE) numbers, we use the same bins as computed for our reliability diagrams, and compute

$$ECE = \sum_{i=1}^{K} P(i) \cdot |o_i - e_i|, \tag{20}$$

where $P(i)$ is the empirical probability a randomly chosen point lies in bin $i$, $o_i$ is the accuracy within bin $i$, and $e_i$ is the average confidence in bin $i$.

We adapted the SWAG authors' implementation at https://github.com/wjmaddox/swa_gaussian to include the ACNML procedure for test time evaluation, and include a copy of the modified codebase in the supplementary materials with instructions on how to reproduce our experiments. We additionally include pretrained models that were used for our experiments. Experiments were conducted using a mix of local GPU servers and Google Cloud Program compute resources.

For the MNIST experiments, we used a feedforward network with 2 hidden layers of size 1200, with no data augmentation. The posterior is factored as independent Gaussians for each parameter, with the prior for each parameter being a zero-mean Gaussian with standard deviation 0.1.

| CIFAR10 Results | VGG16 | | | WideResNet28x10 | | |
|---|---|---|---|---|---|---|
| | NLL | Accuracy | ECE | NLL | Accuracy | ECE |
| ACNML-SWAGD (ours) | $0.2180 \pm 0.0041$ | $93.23 \pm 0.09$ | $0.0246 \pm 0.0010$ | $0.1130 \pm 0.0012$ | $96.38 \pm 0.03$ | $0.0122 \pm 0.0006$ |
| ACNML-KFAC (ours) | $0.2329 \pm 0.0028$ | $93.14 \pm 0.08$ | $0.0361 \pm 0.0016$ | - | - | - |
| MAP (SWA) | $0.2694 \pm 0.0056$ | $93.23 \pm 0.13$ | $0.0430 \pm 0.0010$ | $0.1128 \pm 0.0014$ | $96.41 \pm 0.01$ | $0.0099 \pm 0.0004$ |
| SWAGD | $0.2257 \pm 0.0047$ | $93.31 \pm 0.04$ | $0.0284 \pm 0.0002$ | $0.1125 \pm 0.0012$ | $96.28 \pm 0.04$ | $\mathbf{0.0042} \pm 0.0003$ |
| SWAG | $\mathbf{0.2016} \pm 0.0031$ | $93.60 \pm 0.10$ | $0.0158 \pm 0.0030$ | $0.1122 \pm 0.0009$ | $96.32 \pm 0.08$ | $0.0088 \pm 0.0006$ |
| KFAC-Laplace | $0.2236 \pm 0.0013$ | $92.76 \pm 0.11$ | $\mathbf{0.0097} \pm 0.0005$ | $0.1197 \pm 0.0031$ | $96.23 \pm 0.02$ | $0.0111 \pm 0.0006$ |
| SWA-Dropout | $0.2562 \pm 0.0025$ | $92.85 \pm 0.14$ | $0.0380 \pm 0.0007$ | $0.1111 \pm 0.0024$ | $96.36 \pm 0.09$ | $0.0107 \pm 0.0008$ |
| SWA-Temp | $0.2481 \pm 0.0245$ | $93.61 \pm 0.11$ | $0.0366 \pm 0.0063$ | $\mathbf{0.1064} \pm 0.0004$ | $96.46 \pm 0.04$ | $0.0080 \pm 0.0007$ |
| SGD | $0.3285 \pm 0.0139$ | $93.17 \pm 0.14$ | $0.0483 \pm 0.0022$ | $0.1294 \pm 0.0022$ | $96.41 \pm 0.10$ | $0.0166 \pm 0.0007$ |

**Table 3: In-distribution comparative results** We see that for in-distribution performance, ACNML variants perform comparably to other methods, without large separations between most methods. Results for SWA-Temp and SGD are taken from Maddox et al. (2019).

# B   FURTHER EXPERIMENTAL RESULTS AND COMPARISONS ON CIFAR10

In addition to the comparisons in the main paper, we additionally compare to SWA-Gaussian (SWAG), which uses a more expressive posterior than SWAG-D, and SWA with Monte Carlo Dropout (Gal and Ghahramani, 2015) (SWA-Drop). For reference, we show in-distribution performance of all methods in Table 3. Overall, performance differences between all methods are quite small, and ACNML's conservative predictions do not improve on NLL or ECE over some baselines on in-distribution performance, which is to be expected, since the main aim of our method is produce more calibrated predictions on **out-of-distribution** tasks.

For completeness, we show expanded results on CIFAR10-Corrupted in Figures 6, 7, and 8. With the same architecture, all methods generally have very similar accuracy. ACNML consistently achieves significantly better ECE on the more severe corruptions, and generally comparable or slightly better NLL.

While evaluating MC-Dropout, we found that adding dropout before each layer in VGG16 (labelled VGG16Drop in 7) significantly improved performance on CIFAR10-C. For fair comparisons, we reran all methods with the VGG16Drop architecture as well.

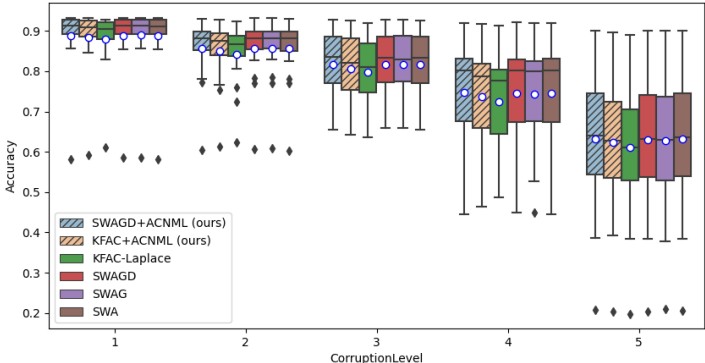

**(a)** CIFAR10C VGG16 Accuracies (higher is better)

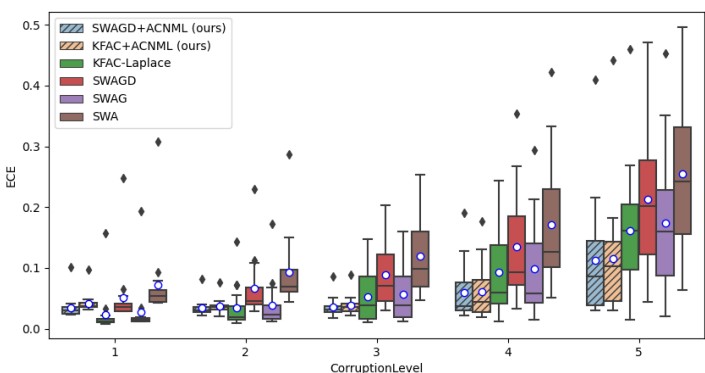

**(b)** CIFAR10C VGG16 ECEs (lower is better)

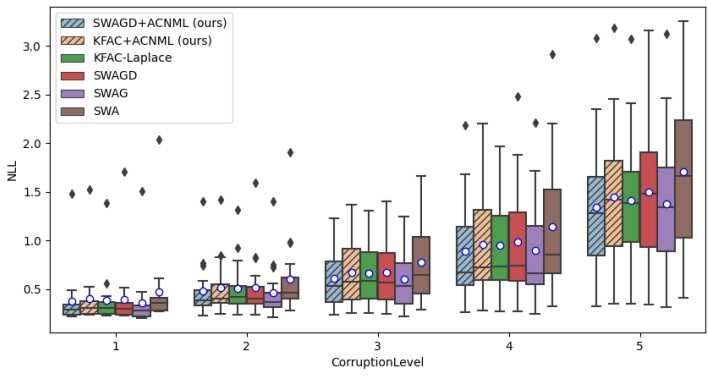

**(c)** CIFAR10C VGG16 NLLs (lower is better)

**Figure 6:** CIFAR10-C performance with the VGG16 architecture. Instantations of our methods are shown in stripes. Boxplots show quartiles of each statistic over all different corruption types of the given intensity, with the mean indicated by a circle. The accuracy (a) and NLL (c) for most methods are similar, but both ACNML variants attain significantly better ECE (b) on the more severe corruptions, as the images move further out of distribution.

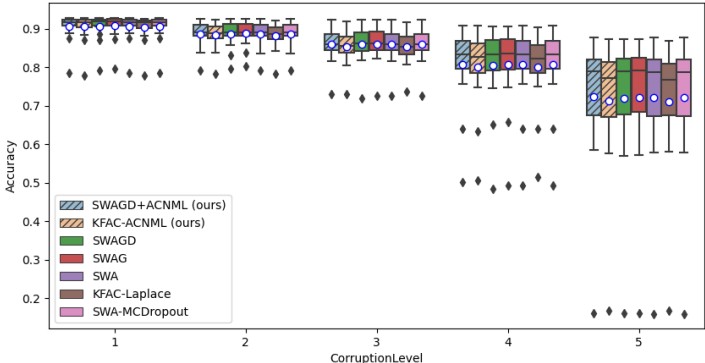

**(a)** CIFAR10C VGG16Drop Accuracies (higher is better)

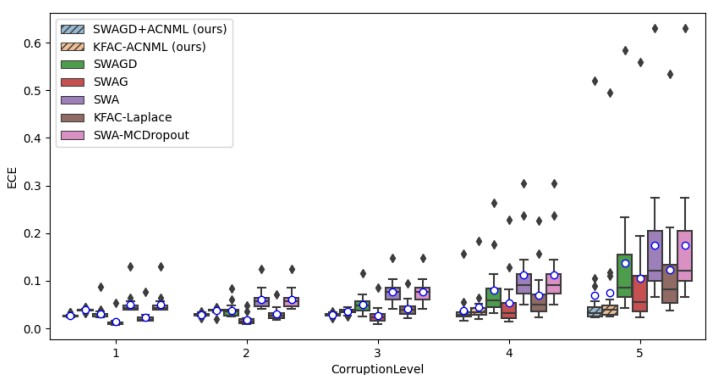

**(b)** CIFAR10C VGG16Drop ECEs (lower is better)

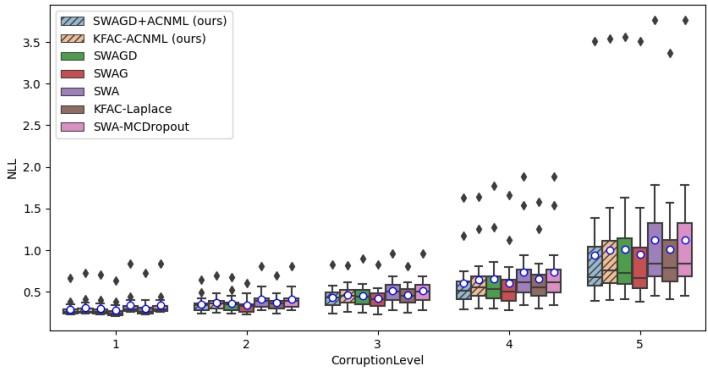

**(c)** CIFAR10C VGG16Drop NLLs (lower is better)

**Figure 7:** CIFAR10-C performance with the VGG16Drop architecture. Instantations of our methods are shown in stripes. Boxplots show quartiles of each statistic over all different corruption types of the given intensity, with the mean indicated by a circle. Again, the accuracy (a) and NLL (c) for most methods are similar, but both ACNML variants attain significantly better ECE (b) on the more severe corruptions, as the images move further out of distribution.

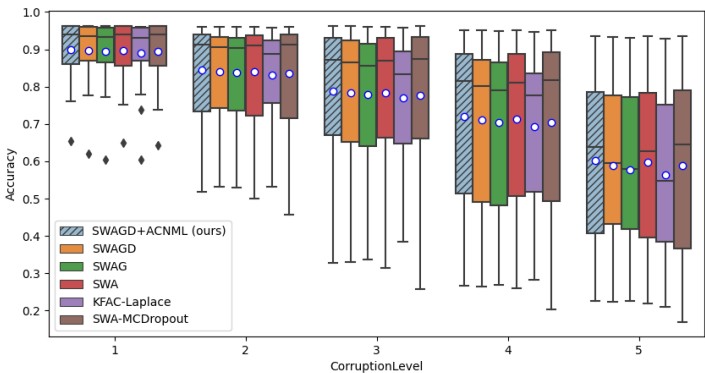

(a) CIFAR10C WRN28x10 Accuracies (higher is better)

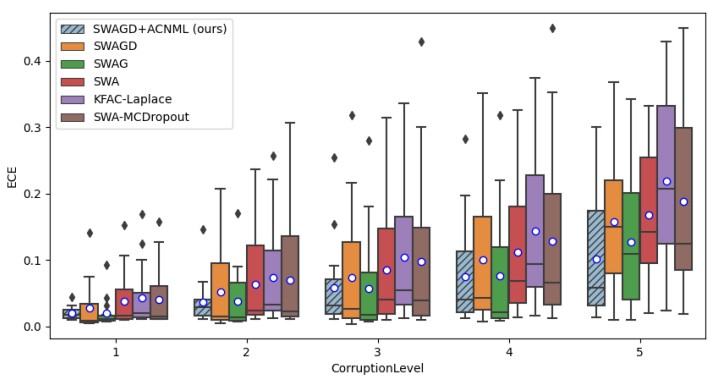

(b) CIFAR10C WRN28x10 ECEs (lower is better)

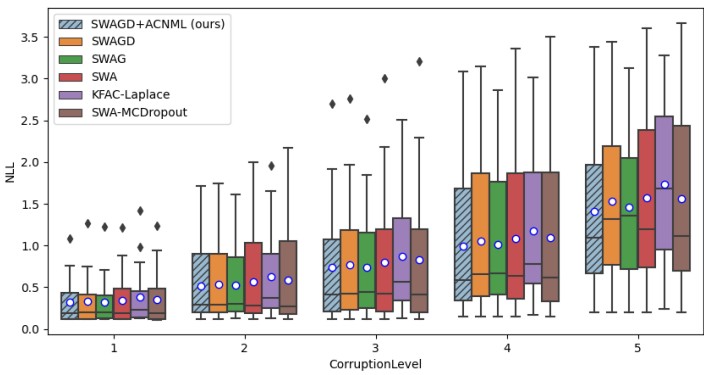

(c) CIFAR10C WRN28x10 NLLs (lower is better)

**Figure 8:** CIFAR10-C performance with the WideResNet28x10 architecture. Instantiations of our methods are shown in stripes. Boxplots show quartiles of each statistic over all different corruption types of the given intensity, with the mean indicated by a circle. Again, we see that ACNML attains better ECE values than comparable methods on the heavier corruptions (b). Note that the best performing prior method, SWAG, uses a substantially more expressive posterior than the diagonal approximation used by SWAGD+ACNML, whereas the comparable SWAGD method attains worse ECE.

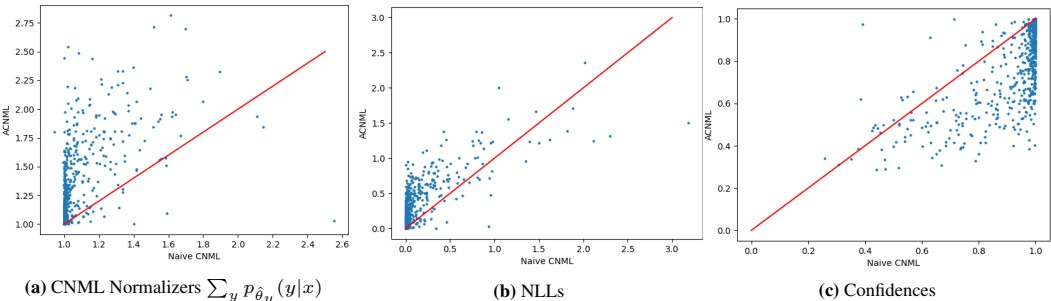

**(a)** CNML Normalizers $\sum_y p_{\hat{\theta}_y}(y|x)$       **(b)** NLLs       **(c)** Confidences

**Figure 9: In Distribution Comparisons between ACNML and naive CNML.** We plot scatter plots of the values of each statistic for naive CNML (x-axis) vs ACNML (y-axis), with the red line indicating Looking at the CNML normalizers, we see that the ACNML adaptation procedure using the approximate posterior is much less constraining than using the training set, resulting in the normalizers being higher for ACNML than naive CNML for almost all inputs. This leads to excess conservatism, with ACNML almost always having lower confidence its predictions, and many inputs with close to 0 NLL with naive CNML having higher NLL with ACNML.

## C  COMPARISONS BETWEEN ACNML AND NAIVE CNML ON MNIST

In this section, we include expanded comparisons between ACNML and a naive implementation of CNML from Bibas et al. (2019b) that computes the MLE/MAP $\hat{\theta}_y$ for each label by appending the query point and label to the dataset and finetuning for $N$ epochs. Both ACNML and naive CNML are initialized from the same MAP solution, with ACNML taking 5 gradient steps on the query point and posterior and naive CNML finetuning with the query point and training set for 5 epochs.

This naive implementation differs slightly from Bibas et al. (2019b) in that we finetune the entire network, while Bibas et al. (2019b) proposed only tuning the last few layers. During the finetuning, we also append the query point and label to every batch in optimization, and downweighting that portion of the loss accordingly to get unbiased gradient estimates. We found this led to more efficient optimization than randomly sampling

We first examine how closely ACNML and naive CNML's predictions match on the same datapoint. To assess this, we compare the CNML normalization terms $\sum_y p_{\hat{\theta}_y}(y|x)$, NLLs, and the confidences of the two methods. The CNML normalization term captures how much each procedure was able to adapt to different labels for that input. A higher normalization term for an input means that we were flexible enough to fit multiple different labels well together with the training set (or approximate posterior in the case of ACNML), and typically means a less confident prediction on that input.

In Figures 9 and 10, We show scatter plots over 1000 randomly selected test points (from the in-distribution test set and the rotated OOD images respectively) comparing the CNML normalizers, NLLs, and confidences of ACNML and naive CNML. In each scatter plot, we include a diagonal red line to illustrate where points would lie if predictions of ACNML and naive CNML matched exactly.

We additionally plot reliability diagrams for MNIST experiments in Figure 11.

For the in-distribution test set, we see from the CNML normalizer plot that the ACNML adaptation procedure using the approximate posterior is much less constraining than using the training set, resulting in the normalizers being higher for ACNML than naive CNML for almost all inputs. This leads to excess conservatism, with ACNML almost always having lower confidence its predictions. As a result, we see that on many points where naive CNML outputted confident correct answers and achieved close to 0 NLL loss, ACNML still incurs some higher losses due to its less confident predictions.

On the OOD rotated images, we again see that ACNML typically adapts more than CNML as measured by the CNML normalizers, though the difference is much less extreme compared to the in-distribution dataset. In the confidence scatter plot, we again see that ACNML tends to make lower confidence predictions than naive CNML (especially when naive CNML's predictions are confident), and as seen in Table 1 and Figure 11, result in ACNML having better NLL and calibration on the OOD inputs.

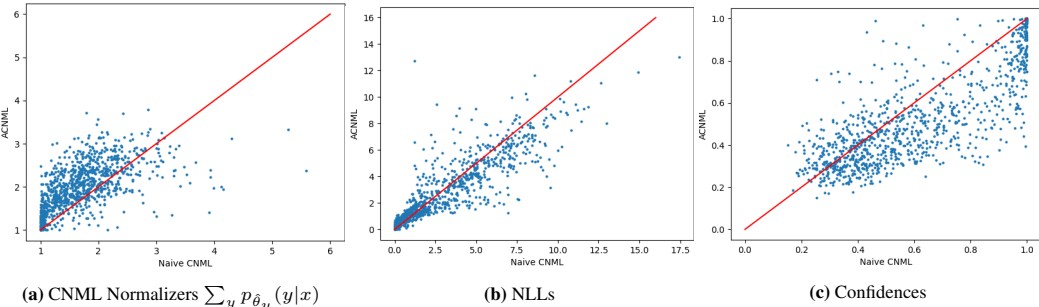

**(a)** CNML Normalizers $\sum_y p_{\hat{\theta}_y}(y|x)$         **(b)** NLLs         **(c)** Confidences

**Figure 10: OOD Comparisons between ACNML and naive CNML.** We plot scatter plots of the values of each statistic for naive CNML (x-axis) vs ACNML (y-axis). Looking at the CNML normalizers, we again see that the ACNML adaptation procedure using the approximate posterior is less constraining than using the training set, with the normalizers being higher for ACNML than naive CNML for most inputs (though to lesser extent than the in-distribution data). ACNML again outputs more conservative predictions with lower confidence on many inputs, which leads to better NLL and calibration on the OOD dataset, unlike with the in-distribution test set.

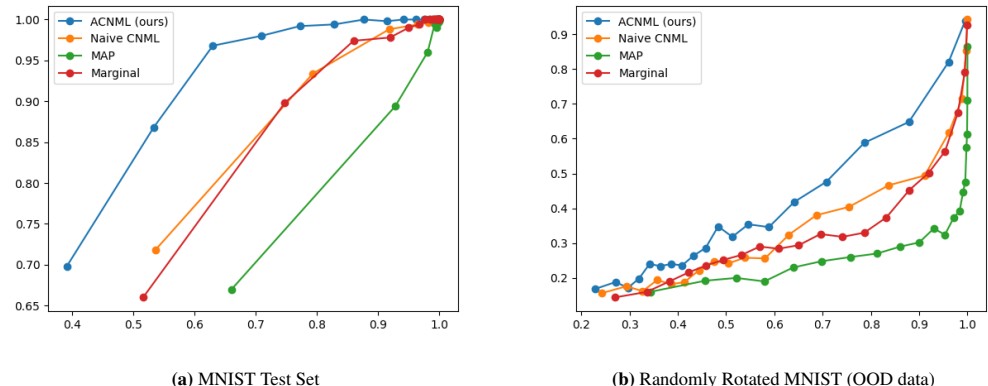

**(a)** MNIST Test Set             **(b)** Randomly Rotated MNIST (OOD data)

**Figure 11:** Reliability diagrams plotting confidence vs. accuracy for Bayes-by-Backprop experiments on the MNIST test set and the randomly rotated MNIST test set (OOD). ACNML's conservative predictions provided better calibrated predictions on the OOD test set.

**Handling multiple MLEs in CNML**: Strictly speaking, the CNML distribution is not well defined when there exist multiple potential MLEs $\hat{\theta}_y$ that can output different predictions (prior references to CNML typically assume such MLEs are unique). However, the non-convexity of the objective for deep neural networks means multiple MLEs can exist, and to properly define CNML in this case, we would need to select a particular MLE to use when assigning probabilities in CNML. In line with the min-max formulation of CNML, we propose to select the MLE $\hat{\theta}_y$ that maximizes the likelihood $p_{\hat{\theta}_y}(y|x)$ of the query point and proposed label, as this is the choice that maximizes the regret for that particular label over all MLEs.

With our naive CNML instantiation, we observe that during the finetuning for each query point $x$ and label $y$, the predicted probability of that label $p_\theta(y|x)$ does not monotonically increase over iterations as we might hope (since we initialize $\theta$ to be the MLE of the training set, then update it to maximize likelihood of the training set with the query point and label), but can potentially oscillate substantially throughout the finetuning process. We suspect this is due to the stochasticity in the optimization procedure from to sampling minibatches of the training data causing the trajectory of parameters can potentially visit several different (approximate) local optima that output different predictions on the query point. While our instantiation of naive CNML simply used the parameter found at the end of 5 epochs, we additionally compare against a variant that explicitly tries to select the MLE that maximizes the likelihood of the proposed label. This variant heuristically uses the bset value of $p_\theta(y|x)$ over all $\theta$ encountered in the last epoch of finetuning. We see in Table 4 that this variant,

| | MNIST | | | Rotated MNIST (OOD) | | |
|---|---|---|---|---|---|---|
| | NLL | Accuracy | ECE | NLL | Accuracy | ECE |
| ACNML (ours) | $0.1818 \pm 0.0032$ | $97.28 \pm 0.21$ | $0.1013 \pm 0.0006$ | $\mathbf{2.766} \pm 0.0197$ | $37.34 \pm 0.06$ | $\mathbf{0.1540} \pm 0.0023$ |
| MAP | $0.0864 \pm 0.0025$ | $97.28 \pm 0.21$ | $\mathbf{0.0047} \pm 0.0006$ | $3.994 \pm 0.072$ | $37.29 \pm 0.02$ | $0.4371 \pm 0.0094$ |
| Marginal | $0.1069 \pm 0.0067$ | $97.22 \pm 0.24$ | $0.0313 \pm 0.0010$ | $3.017 \pm 0.022$ | $37.63 \pm 0.31$ | $0.2928 \pm 0.0032$ |
| naive CNML | $\mathbf{0.0774} \pm 0.0024$ | $\mathbf{98.05} \pm 0.08$ | $0.0231 \pm 0.0001$ | $3.100 \pm 0.057$ | $37.33 \pm 0.34$ | $0.2497 \pm 0.0072$ |
| naive CNML (max over itrs) | $0.0882 \pm 0.0018$ | $97.90 \pm 0.23$ | $0.0355 \pm 0.0005$ | $2.991 \pm 0.021$ | $37.34 \pm 0.003$ | $0.1858 \pm 0.0075$ |

**Table 4:** Expanded comparative results for ACNML on MNIST using a posterior obtained via Bayes by Backprop.

denoted naive CNML (max over itrs), gives more conservative predictions than naive CNML and improves in NLL and calibration on the OOD dataset.

## D    NMAP AND ACNML

NML type methods can be extended with a prior-like regularization term on the selected parameter, resulting in Normalized Maximum a Posteriori (NMAP)(Kakade et al., 2006), also referred to as Luckiness NML (Grunwald, 2004). For a regularizer given by $\log p(\theta)$, NMAP assigns probabilities according to

$$p^{\text{NMAP}}(x^n) \propto p_{\hat{\theta}(x^n)}(x^n) \qquad\qquad \hat{\theta}(x^n) = \underset{\theta}{\text{argmax}} \log p_\theta(x^n) + \log p(\theta).$$

Similarly to CNML, there are several variations on NMAP or LNML that predict slightly different distributions, but we adopt the one of the same form as our CNML. Similarly to how NML was extended to CNML, NMAP can be extended to a conditional version, again with the $\hat{\theta}$'s being chosen via MAP rather than MLE. As mentioned in Section 3.1, with a non-uniform prior, ACNML actually approximates a version of conditional NMAP, with the Bayesian prior term on the parameters corresponding to the additional regularizer.

We also note that with the calculations in section 3.1, we see that CNML can be viewed as performing NMAP on the new test point, with a regularizer corresponding to the likelihoods on the training data. In this perspective, ACNML approximates CNML by using an approximation to that training loss regularizer.

## E    DETAILS OF ANALYSIS IN SECTION 3.2

### E.1    BOUNDING ERROR IN PARAMETER ESTIMATION

Here we state the primary theorem of Giordano et al. (2019) along with the necessary definitions and assumptions.

Here, we attempt to estimate an unknown parameter $\theta \in \Omega_\theta \subseteq \mathbb{R}^D$ where $\Omega_\theta$ is compact. Suppose we have a dataset $N$ datapoints and a weight vector $w_1, \ldots, w_N$. Let $g_i(\theta)$ denote the gradient of the loss at datapoint $i$ evaluated at $\theta$, and $h_i(\theta)$ the Hessian. We can then define

$$G(\theta, w) = \frac{1}{N} \sum_{i=1}^{N} w_i g_i(\theta) \tag{21}$$

$$H(\theta, w) = \frac{1}{N} \sum_{i=1}^{N} w_i h_i(\theta). \tag{22}$$

The MLE $\hat{\theta}(w)$ for the dataset weighted by $w$ is given by solving for $G(\hat{\theta}(w), w) = 0$. Let $1_w$ denote the vector of weights consisting of all 1s. We define $\hat{\theta}_1$ to be the MLE for the whole unweighted dataset, which is equivalent to evaluating $\hat{\theta}(1_w)$ and also define the corresponding Hessian $H_1 = H(\hat{\theta}_1, 1_w)$. We now wish to estimate $\hat{\theta}(w)$ using a first order approximation around $\hat{\theta}_1$ given by

$$\hat{\theta}_{\text{IJ}}(w) = \hat{\theta}_1 - H_1^{-1} G(\hat{\theta}_1, \Delta w), \tag{23}$$

where we define $\Delta_w = w - 1_w$. The theorem will proceed to bound $\left\| \hat{\theta}(w) - \hat{\theta}_{\mathrm{IJ}} \right\|_2$ for suitable weights $w$.

Now we further define $g(\theta) \in \mathbb{R}^{N \times D}$ to be the concatenation of all $g_i(\theta)$s and similarly for $h(\theta) \in \mathbb{R}^{N \times D \times D}$. We let $\|g(\theta)\|_p$ and $\|h(\theta)\|_p$ to refer to the $p$-norms when treating those as vector quantities.

**Assumption 1** (Smoothness): For all $\theta \in \Omega_\theta$ each $g_n(\theta)$ is continuously differentiable.

**Assumption 2** (Non-degeneracy): For all $\theta \in \Omega_\theta$, $H(\theta, 1_w)$ is nonsingular and

$$\sup_{\theta \in \Omega_\theta} \left\| H(\theta, 1_w)^{-1} \right\|_{op} \leq C_{op} \leq \infty. \tag{24}$$

**Assumption 3** (Bounded averages): There exist finite constants $C_g$ and $C_h$ such that $\sup_{\theta \in \Omega_\theta} \frac{1}{\sqrt{N}} \|g(\theta)\|_2 \leq C_g$ and $\sup_{\theta \in \Omega_\theta} \frac{1}{\sqrt{N}} \|h(\theta)\|_2 \leq C_h$.

**Assumption 4** (Local Smoothness): There exists a $\Delta_\theta > 0$ and a finite constant $L_h$ such that $\left\| \theta - \hat{\theta}_1 \right\|_2 \leq \Delta_\theta$ implies $\frac{\|h(\theta) - h(\hat{\theta}_1)\|_2}{\sqrt{N}} \leq L_h \left\| \theta - \hat{\theta}_1 \right\|_2$.

**Assumption 5** (Bounded weight averages). $\frac{1}{\sqrt{N}} \|w\|_2$ is uniformly bounded for all $w \in W$ by a finite constant $C_w$.

We note that assumption 2 is equivalent to $H_1$ being strongly positive definite. Assumption 5 is not relevant for our use cases, but is stated for completeness.

**Condition 1** (Set Complexity): There exists a $\delta \geq 0$ and corresponding set $W_\delta \subseteq W$ such that

$$\max_{w \in W_\delta} \sup_{\theta \in \Omega_\theta} \left\| \frac{1}{N} \sum_{i=1}^{N} (w_i - 1) g_i(\theta) \right\|_1 \leq \delta. \tag{25}$$

$$\max_{w \in W_\delta} \sup_{\theta \in \Omega_\theta} \left\| \frac{1}{N} \sum_{i=1}^{N} (w_i - 1) h_i(\theta) \right\|_1 \leq \delta. \tag{26}$$

Condition 1 essentially describes the set of weight vectors for which $\hat{\theta}_{\mathrm{IJ}}$ will be an accurate approximation within order $\delta$.

**Definition 1**: Given assumptions 1-5, define

$$C_{\mathrm{IJ}} = 1 + D C_w L_h C_{op} \tag{27}$$

$$\Delta_\delta = \min\{\Delta_\theta C_{op}^{-1}, \frac{1}{n} C_{\mathrm{IJ}}^{-1} C_{op}^{-1}\}. \tag{28}$$

We now state the main theorem of Giordano et al. (2019).

**Theorem** (Error Bound for the approximation). Under assumptions 1-5 and condition 1,

$$\delta \leq \Delta_\delta \Rightarrow \max_{w \in W_\delta} \left\| \hat{\theta}_{\mathrm{IJ}}(w) - \hat{\theta}(w) \right\|_2 \leq 2 C_{op}^2 C_{\mathrm{IJ}} \delta^2. \tag{29}$$

We can now apply the above theorem to provide error bounds for a setting where we have a training set of $n$ datapoints and wish to consider the MLE after adding a new datapoint $z$. The issue is that the theorem as stated bounds the error of the approximation when the approximation is centered around the uniform weighting over all the datapoints, which would be appropriate for considering the impact of *removing* datapoints from the dataset.

To apply the theorem to bound the effects of *adding* a datapoint, we have to do some slight manipulation. We apply the previous theorem with $N = n + 2$, where $g_i(\theta)$ correspond to the gradients of training data point $i$ for $i$ in $(1, \ldots, n)$, $g_{n+1} = -\nabla \log p_\theta(z)$, and $g_{n+2} = \nabla \log p_\theta(z)$, and similarly for the Hessians $h_i(\theta)$. We have thus added the query point to the dataset, as well as another fake point that serves to cancel out the contribution of the query point under a uniform weighting, so $G(\theta, 1_w)$ and $H(\theta, 1_w)$ are the mean gradients and Hessians for just the training set. Now supposing

assumptions 1-5 are met for this problem, then we need to check condition 1 for the particular $W_\delta$ that contains the vector $\bar{w}$ of all 1s, except for a 2 in the last entry. We can then find the smallest $\delta$ that satisfies

$$\sup_{\theta \in \Omega_\theta} \left\| \frac{1}{N+2} g_{n+2}(\theta) \right\|_1 \leq \delta \tag{30}$$

$$\sup_{\theta \in \Omega_\theta} \left\| \frac{1}{N+2} h_{n+2}(\theta) \right\|_1 \leq \delta, \tag{31}$$

and so long as $\delta \leq \Delta_\delta$, applying the theorem bounds $\left\| \hat{\theta}_{\mathrm{IJ}}(\bar{w}) - \hat{\theta}(\bar{w}) \right\|_2$.

**Commentary**: The above theorem gives explicit conditions for the accuracy of the approximation that we can verify for a particular training set and query point. Under assumptions that we have some limiting procedure for growing the training set such that the constants defined hold uniformly, we can extend this to an asymptotic statement to explicitly say that the approximation error decays as $O(n^{-2})$.

### E.2 Bounding error in the resulting CNML distribution

We now provide the proof for Proposition 3.2, which we restate here. For notational simplicity, we ignore any dependence on the input $x$, which we consider fixed.

**Proposition E.1** (3.2). *Suppose $z \in \mathcal{Z}$ with $|\mathcal{Z}| = k$ (for example classification with $k$ classes). Let $\hat{\theta}_z$ be the exact MLE after appending $z$ to the training set, and let $\tilde{\theta}_z$ be an approximate MLE with $\left\| \hat{\theta}_z - \tilde{\theta}_z \right\| \leq \delta$ for all $z$. Further suppose $\log p_\theta(z)$ is $L$-Lipschitz in $\theta$.*

*Denote the exact CNML distribution $p_{CNML}(z) \propto p_{\hat{\theta}_z}(z)$ and an approximate CNML distribution $p_{ACNML}(z) \propto p_{\tilde{\theta}_z}(z)$. Then, we have the bound*

$$\sup_z |\log p_{CNML}(z) - \log p_{ACNML}(z)| \leq 2L\delta. \tag{32}$$

*Proof.* The assumed bound $\left\| \hat{\theta}_z - \tilde{\theta}_z \right\|_2 \leq \delta$ combined with $L$-Lipschitzness implies a bound on differences of logits of each class

$$\left| \log p_{\hat{\theta}_z}(z) - \log p_{\hat{\theta}_z}(z) \right| \leq L\delta. \tag{33}$$

We note that the log probabilities of the exact CNML distribution $p_{CNML}$ ($p_{ACNML}$ is given by a similar expression using $\tilde{\theta}_z$ instead of $\hat{\theta}_z$) is given by

$$\log p_{CNML}(z) = \log p_{\hat{\theta}_z}(z) - \log \sum_{z' \in \mathcal{Z}} p_{\hat{\theta}_{z'}}(z'). \tag{34}$$

For any $z \in \mathcal{Z}$, we can then expand, apply the triangle inequality and then Equation 33 to obtain

$$|\log p_{CNML}(z) - \log p_{ACNML}(z)| = \left| \log p_{\hat{\theta}_z}(z) - \log p_{\tilde{\theta}_z}(z) - \log \sum_{z' \in \mathcal{Z}} p_{\hat{\theta}_{z'}}(z') + \log \sum_{z' \in \mathcal{Z}} p_{\tilde{\theta}_{z'}}(z') \right| \tag{35}$$

$$\leq \left| \log p_{\hat{\theta}_z}(z) - \log p_{\tilde{\theta}_z}(z) \right| + \left| \log \sum_{z' \in \mathcal{Z}} p_{\hat{\theta}_{z'}}(z') - \log \sum_{z' \in \mathcal{Z}} p_{\tilde{\theta}_{z'}}(z') \right| \tag{36}$$

$$\leq L\delta + \left| \log \sum_{z' \in \mathcal{Z}} p_{\hat{\theta}_{z'}}(z') - \log \sum_{z' \in \mathcal{Z}} p_{\tilde{\theta}_{z'}}(z') \right|. \tag{37}$$

We now bound the difference between the log-normalizers $\left| \log \sum_{z'} p_{\hat{\theta}_{z'}}(z') - \log \sum_{z'} p_{\tilde{\theta}_{z'}}(z') \right|$.

We first let $p_{\min}(z) = \min\{p_{\hat{\theta}_z}(z), p_{\tilde{\theta}_z}(z)\}$ and $p_{\max}(z) = \max\{p_{\hat{\theta}_z}(z), p_{\tilde{\theta}_z}(z)\}$, and note that Equation 33 implies $\log p_{\max}(z) \leq \log p_{\min}(z) + L\delta$ for all $z$. We then bound the difference in log-normalizers

$$\left| \log \sum_{z' \in \mathcal{Z}} p_{\hat{\theta}_{z'}}(z') - \log \sum_{z' \in \mathcal{Z}} p_{\tilde{\theta}_{z'}}(z') \right| \leq \log \sum_{z' \in \mathcal{Z}} p_{\max}(z') - \log \sum_{z' \in \mathcal{Z}} p_{\min}(z') \tag{38}$$

$$= \log \frac{\sum_{z' \in \mathcal{Z}} p_{\max}(z')}{\sum_{z' \in \mathcal{Z}} p_{\min}(z')} \tag{39}$$

$$= \log \frac{\sum_{z' \in \mathcal{Z}} \exp(\log p_{\max}(z'))}{\sum_{z' \in \mathcal{Z}} p_{\min}(z')} \tag{40}$$

$$\leq \log \frac{\sum_{z' \in \mathcal{Z}} \exp(\log p_{\min}(z') + L\delta)}{\sum_{z' \in \mathcal{Z}} p_{\min}(z')} \tag{41}$$

$$= \log \frac{\exp(L\delta) \sum_{z' \in \mathcal{Z}} p_{\min}(z')}{\sum_{z' \in \mathcal{Z}} p_{\min}(z')} \tag{42}$$

$$= L\delta. \tag{43}$$

Plugging back into Equation 37, we have the following bound for all $z \in \mathcal{Z}$

$$|\log p_{\mathrm{CNML}}(z) - \log p_{\mathrm{ACNML}}(z)| \leq 2L\delta. \tag{44}$$

$\square$

