# OpenReview forum: "Amortized Conditional Normalized Maximum Likelihood"
_ICLR.cc/2021/Conference — Reject_

### Official Review · AnonReviewer1 · 2020-10-26
**a paper that may need careful checking before submission**

**Rating:** 5
**Confidence:** 3

**Review:**

This paper proposes ACML method which provides a tractable approximation to the CNML distribution and reduces the computational complexity of CNML method.

Strength:
1. Provide a tractable approximation for CNML which makes training easier and faster.
2. Theoretically prove the approximation error is bounded, but the proof part has too many typos and cannot be confirmed.

Weakness:
1. The author didn’t use the experiment to show the accuracy of CNML distribution and ACNML distribution. The author only compared the training time of these two methods. Since in equation 18, the given bound of the approximation error increases with k, we don’t know whether this approximation is well bounded.
2. In Table 1, MAP is a better method than ACNML for the MNIST dataset.
3. In Table 3, it misses the results of ACNML-KFAC for WideResNet28x10 dataset, and it seems that the proposed methods don’t outperform other existing methods.

Typos:
1. In Equation 34, there should be a θ ̃_z instead of θ ̂_z
2. In Equation 36, the third and fourth terms sum over z’, but there is no such parameter, and the symbol of absolute value disappears, due to this reason, I cannot verify if this equation holds.
3. Looks like Equation 39 misses a minus symbol.

---

> ### Author Response · Authors · 2020-11-15
> **Response to Reviewer 1**
>
> We thank the reviewer for helpful feedback. Based on the review, we believe the reviewer’s main concerns about the paper are the typos in the proof of our bounds on approximation error (Proposition 3.2), the dependence in the bound on the number of labels $k$, and that our method does not improve performance in terms of accuracy for in-distribution data points.
>
> **Fixed typos and improved theoretical bound:** In the updated text, we have fixed the typos and improved the clarity in the proof of Proposition 3.2 (Appendix D.2). We hope the correctness can now be confirmed by the reviewer. Additionally, we have improved the bound in Prop 3.2 to remove the dependence on $k$, the number of labels.
>
> **Regarding in-distribution empirical performance:** We emphasize that the goal of our method is to provide better uncertainty estimates on OOD inputs, rather than improving in-distribution performance. We believe this is an important problem to tackle, as typical deep neural nets tend to make overconfident incorrect predictions on OOD inputs, while ACNML’s calibrated predictions provide much more conservative uncertainty estimates for unfamiliar, out-of-distribution inputs where errors are more likely. We have edited the introduction to clarify our goal is to obtain reliable uncertainty estimates on OOD inputs and motivate why reliable uncertainty estimation is important.
>
> While the conservativeness of ACNML does result in somewhat worse NLL and ECE on the in-distribution MNIST test set, ACNML does maintain the same accuracy (0/1 loss) as other methods, and also maintains competitive performance in all metrics (in Table 3) on the in-distribution CIFAR10 test set. We emphasize again that improving in-distribution performance is not at all the goal of our method, as we simply wish to show that ACNML can provide better uncertainty estimates on OOD inputs (Figure 4 and 5 in the updated text, Figures 1 and 2 in the original submission) without significantly compromising in-distribution performance.
>
> **Empirically measuring accuracy of the ACNML approximation:** To help assess the accuracy of the approximation ACNML uses, we are working on adding additional comparisons with the much more computationally expensive method used by Bibas et al, which approximates CNML by appending the new query point/label to the training set and finetuning for several epochs to compute new parameters. We are working to compare the two on the simpler MNIST domain, where the computational costs are a bit less prohibitive.

---

> ### Author Response · Authors · 2020-11-20
> **Followup to Reviewer 1**
>
> We would like to again ask the reviewer to respond as to whether their concerns in the paper have been addressed in our responses and updated manuscript. Please let us know if any concerns or questions remain.

---

### Official Review · AnonReviewer4 · 2020-10-29
**An interesting new direction for computing well-calibrated predictions**

**Rating:** 6
**Confidence:** 3

**Review:**

This paper explores the application of an alternative inference scheme for learning well-calibrated predictive models especially target to ut-of-distribution samples. The paper is based on the so-called conditional normalized maximum likelihood, which is an inference approach based on min-max regret principles. Experimental evaluation shows promising performance to robust predictions under distributional shifts.

Strong points:

- The paper explores the use of alternative and well-founded inference approaches.

- The presented solution is well-motivated and scales to large data samples and large models.

- The empirical evaluation shows promising results on ut-of-distribution samples.

Weak points:
- The adaptation of the conditional normalized maximum likelihood approach presented in this work lacks the min-max regret guarantees provided by the original framework.

- Empirical evidence about the advantage of the method is limited.

- The presented method is too cautious when making predictions which are not out-of-distribution samples, and its accuracy is not competitive in this regime.

I support the recommendation for acceptance of this work because it explores the use of a well-founded inference framework, the conditional normalized maximum likelihood, which optimal min-max regret guarantees but which is computationally intractable in its exact form. This work proposes an approximate inference scheme which makes the application of this inference framework tractable for deep neural networks.


Questions:

- Does  the provided analysis of second-order approximations provide any relevant insight? It is based on very strong assumptions which do not hold on reality.

- What happens when there are multiple maximum-likelihood estimates like happens in deep ensembles? Every member of the ensemble achieves zero training error.

- Why don't you compare with Bibas et al. (2019)?

Minor comments:
-  Space below Figure 1.

- Y label Figure 1 b) and c) is not readable.

- Missing Eq. reference in Pag. 11

- Figure 2 is hard to read. Please consider another format (bar based plot?).

Post-rebuttal: I thanks the authors for their answers to my concerns. After looking the reviews of the other reviewers and the ongoing discussions, I have decided to keep my score. I think this paper makes a novel proposal which deserves to be published.

---

> ### Author Response · Authors · 2020-11-15
> **Response to Reviewer 4**
>
> We thank the reviewer for the helpful comments. We have updated the paper to clarify the insights from the theoretical analysis section and that ACNML maintains competitive performance on in-distribution inputs despite its more cautious predictions.
>
> *\> The presented method is too cautious when making predictions which are not out-of-distribution samples, and its accuracy is not competitive in this regime.*
> While ACNML does make cautious predictions on in-distribution samples, our results indicate it is still competitive in terms of accuracy (0/1 loss). In the CIFAR10 test results in Table 3 (Appendix B), we show that the accuracy of ACNML-SWAGD is almost exactly the same as with the deterministic baseline SWA. With VGG16, ACNML-KFAC does have slightly lower accuracy than SWA (by 0.1%, which is within the error deviations across seeds), but has higher accuracy than the corresponding Bayesian marginalization method using KFAC-Laplace.
>
> Similarly on MNIST, all methods achieve almost identical accuracy in-distribution. ACNML does perform worse in terms of NLL on the in-distribution MNIST test set, where all methods achieve very high accuracy and the cautious predictions are especially detrimental. However, on CIFAR10 with VGG16, ACNML variants provide overall competitive NLLs (better than the deterministic SWA baseline) while slightly worse than the best performing Bayesian methods. With the WideResNet28x10 architecture where accuracies are overall higher, ACNML is still competitive with other methods, with the NLLs of ACNML, SWA, and the Bayesian methods SWAG and SWAG-D all within margins of error of one another.
>
> Overall, our empirical results indicate ACNML’s conservative predictions provide *better calibration on OOD inputs* while maintaining *overall competitive performance on in-distribution inputs*. We have clarified in the experiments section that ACNML does retain competitive performance on the in-distribution CIFAR10 test set.
>
> **Reviewer Questions**
> **Comparison to Bibas et al:** We initially did not include comparisons with Bibas et al due to the prohibitive computational costs involved. The inference time included in table 2 for direct CNML was only for finetuning with a single epoch for each label, while Bibas et al finetune for 10 epochs, which would incur even higher computational costs. We are working to add comparisons with Bibas et al on the simpler MNIST domains, where computational costs are less prohibitive. Even on MNIST, this comparison takes a very large amount of time to run (at test time, it takes 1-2 minutes to evaluate a single input) due to the computational cost of this prior method, but we will provide results as soon as it completes.
>
> **Insights from theoretical analysis:** We believe the theoretical section 3.2 helps provide an understanding of why utilizing Gaussian approximate posteriors in ACNML should be expected to give a reasonable approximation to CNML for large datasets by drawing connections to influence functions. While we agree that the assumptions required to formalize this do not hold for deep neural networks, these assumptions are discussed in Giordano et al (2019) as being realistic for many commonly used statistical models. Additionally, as we mention in the paper, influence function calculations based on these second-order approximations have been shown in Koh and Liang (2017) to be practically useful for estimating how individual datapoints affect deep neural net predictions, despite the assumptions needed being too restrictive to hold deep learning.
>
> **Multiple MLEs:** From our understanding, the reviewer’s question regarding multiple MLEs is asking how CNML is defined when there are multiple different parameters $\hat \theta_y$ that maximize the likelihood of the training set and the new query point/label.
>
> If we append the query point $(x, y)$ to a dataset and find multiple MLEs that achieve 0 loss, then all the models correctly predict label y with probability ~1, so having multiple MLEs does not really matter in this scenario, as CNML only cares about the query point prediction from the new MLE rather than the parameters itself.
>
> However, in general, if the multiple MLEs do not all achieve 0 loss, then they can have different predictions on the query point. Here, CNML is not well defined, as selecting different $\hat \theta_y$’s would result in different CNML distributions. We are not aware of any past discussion of this issue in CNML, as it is typically assumed that the MLEs are unique.. To properly define CNML in such ambiguous cases, we could perhaps choose the $\hat \theta_y$ that assigns the highest probability to the proposed label $y$ on the query point, which would make sense in using the worst-case regret over different MLEs and labels.
>
> We hope these answer the reviewer’s questions.

---

### Official Review · AnonReviewer3 · 2020-10-29
**Good paper overall; however the proposed method is straight-forward and not that convincing**

**Rating:** 6
**Confidence:** 3

**Review:**

The paper presents an approach based on conditional normalized maximum likelihood (CNML) for uncertainty estimation, calibration, and out-of-distribution robustness with deep networks. CNML is intractable to compute in general and therefore the authors propose a tractable approximation which uses approximate Bayesian inference techniques. Experimentally, the authors show that their new approach is competitive and sometimes better than existing approaches for uncertainty estimation and calibration on out-of-distribution test data points.

The paper is well written, easy to understand and the authors are well versed with the domain (the writing clearly demonstrates scholarship, knowledge and expertise). As far as I can tell, related work is appropriately cited and the authors have done an excellent job in describing how their current work advances and relates to prior work on this topic. My criticisms/suggestions which I outline next are purely constructive and my hope is that they will greatly improve the paper:

1. The algorithmic advances (Algorithm 1) are not that novel. I am not sure why the equation for the regret and definitions of unconditional normalized maximum likelihood are presented in the paper. The authors can simply start with the non-regret based definition of CNML given in Equation 6 and derive their approximation (the minimax optimization is not relevant to the algorithm).
2. It would be informative to include comparison of your method with the method of Bibas et al. 2019 in terms of accuracy, NLL and ECE scores (at least for the MNIST dataset). In a way, what you are proposing is a faster, more scalable approach and it would be informative to know how far its estimates are from a much slower (but potentially more accurate) approach by Bibas et al.
3. Another important comparison missing in the paper is how various methods perform on synthetic datasets where CNML can be exactly/accurately computed. As such, it is unclear whether smoothing/averaging effects of CNML are responsible for performance improvements on out-of-distribution samples or whether it is highly dependent on the quality of the approximate Bayesian inference method used.
4. Latex tricks are used to somehow squeeze Figure 1 in the paper. Perhaps, what you could have done is removed definitions of (unconditional) NML.
5. Section 3.2, although important from a theoretical point of view does not advance the main thesis/point of the paper (a faster approach for uncertainty estimation/calibration that uses approximate Bayesian inference). Also, most of the results described in section 3.2 are already known (specifically, Theorem 3.1).

Overall, a good paper which advances the state-of-the-art on an important problem (uncertainty estimation and calibration) in neural networks. However, the paper lacks novelty and its presentation and experimental evaluation can be improved.

---

> ### Author Response · Authors · 2020-11-15
> **Response to Reviewer 3**
>
> We thank the reviewer for the helpful review and suggestions. In incorporating the reviewer’s suggestion, we have added a synthetic example where CNML is exactly computable, edited the discussion of the theoretical section to emphasize how it helps us understand the approximation ACNML uses, and are working to add comparisons with the method of Bibas et al.
>
> **Regarding Novelty:** While our proposed algorithm is a straightforward modification of CNML, our contribution is novel in that no previous work has proposed an approximation like this for CNML. We have shown that without our contributions, a naive implementation of CNML is impractically computationally expensive to instantiate in deep learning settings. We are the first to show that an inference procedure based on CNML can be utilized as a computationally practical tool for improving uncertainty estimation and calibration on OOD inputs with deep neural networks.
>
> **Adding Comparisons with Bibas et al:** We agree with the reviewer that it would be informative to compare ACNML against the finetuning method of Bibas et al, and are working to add comparisons on the MNIST settings, where computational costs are a bit less prohibitive. This comparison takes a very large amount of time to run (at test time, it takes 1-2 minutes to evaluate a single input even on MNIST), due to the computational cost of this prior method, but we will provide results as soon as it completes.
>
> **Added Synthetic Example:** We have added an illustrative example of CNML on a synthetic 2-dimensional logistic regression task (shown in Figures 1-3), where we can exactly compute CNML. This allows us to illustrate how CNML provides conservative predictions for out-of-distribution inputs.
>
> **Theory:** While utilizing approximate Gaussian posteriors enables faster computation in our method, it is not obvious it actually allows us to approximate CNML well. We believe the theoretical section 3.2 helps provide understanding and intuition as to why this approximation should be reasonably accurate by drawing connections to the concept of influence functions from classical statistics, with the presented theorem formalizing the intuition and providing guarantees on the accuracy of the approximation under some more restrictive assumptions. Even though the conditions of the theorem do not necessarily hold in deep learning, approximations based on influence functions have been found to be useful for deep learning (as in Koh and Liang). We have edited the analysis section 3.2 to clarify its role in the paper.

---

### Official Review · AnonReviewer2 · 2020-10-29
**paper 2062 Review**

**Rating:** 5
**Confidence:** 3

**Review:**

The authors propose the amortized conditional normalized maximum likelihood (ACNML) method for scalable, general-purpose uncertainty estimation, calibration and out-of-distribution robustness for deep neural networks. ACNML is a tractable approximation to CNML that leverages Bayesian inference by using an approximate posterior in place of optimizing over the dataset during inference.

Based in the formulation provided CNML does not seem to be intractable but prohibitive.

The proposed approach is very simple and intuitive, even after the modification in (19). The analysis of the second-order approximation in Section 3.2 though interesting does not seem to impact the proposed method (as implemented) and the experimental results.

It is difficult to evaluate the advantage of the proposed method because from the results: 1) all compared methods perform about the same in terms of accuracy, 2) the predictions from the proposed models do not necessarily result in better NLL values (as a proxy for uncertainty estimation performance), or calibration on in-distribution datasets. It is only in terms of calibration of OOD calibration where the proposed approach seems to have an edge over existing methods.

Minor:
- $p_\theta(x)$ is not defined.
- there is a missing reference in the Appendix, above (20).

---

> ### Author Response · Authors · 2020-11-15
> **Response to Reviewer 2**
>
> We thank the reviewer for the helpful comments. We agree with the reviewer that CNML is prohibitively computationally expensive to compute rather than necessarily intractable, and have edited the text accordingly, as well fixing the minor issues raised. Based on the review, we believe the reviewer’s main concern is about the benefits of ACNML over prior methods, which we address below.
>
> **Advantages of ACNML in Uncertainty Estimation for OOD Inputs**: We emphasize that our goal with ACNML is not to provide better accuracy or in-distribution performance, but to provide better uncertainty estimation on OOD inputs, which we measure via calibration. We believe that obtaining reliable uncertainty estimates on OOD inputs is an important problem to tackle, as typical deep neural nets tend to make overconfident incorrect predictions on OOD inputs. On the other hand, ACNML successfully provides much better calibrated predictions on OOD inputs (see figures 4 and 5 in the updated text, 1 and 2 in the old), so even though we do not improve in accuracy, the improved calibration means we can at least use the ACNML’s confidence estimates to identify inputs for which we are unable to make accurate predictions. We have edited the introduction to clarify our goal is to obtain reliable uncertainty estimates on OOD inputs and motivate why reliable uncertainty estimation is important.
>
> With regards to using calibration (ECE) vs negative log-likelihood as a measure of uncertainty, prior work in benchmarking uncertainty on OOD inputs (Ovadia et al 2019) primarily use ECE over NLL as the metric by which to compare uncertainty estimates on corrupted image domains like CIFAR10-C.
>
> Citations:
> Y. Ovadia, E. Fertig, J. Ren, Z. Nado, D. Sculley, S. Nowozin, J. V. Dillon, B. Lakshminarayanan,
> and J. Snoek. Can you trust your model’s uncertainty? evaluating predictive uncertainty under
> dataset shift, 2019.

---

> ### Author Response · Authors · 2020-11-20
> **Followup to Reviewer 2**
>
> We would like to again ask the reviewer to respond as to whether their concerns in the paper have been addressed in our responses and updated manuscript. Please let us know if any concerns or questions remain.

---

### Public Comment · ~Zhisheng_Xiao1 · 2020-11-16
**Work that has some high-level connections**

Hi authors. I enjoy reading your manuscript.  During the reading, I realize this work shares some similarities in high-level ideas with our recent work Likelihood Regret: An Out-of-Distribution Detection Score For Variational Auto-encoder (https://papers.nips.cc/paper/2020/hash/eddea82ad2755b24c4e168c5fc2ebd40-Abstract.html).

In particular, our work targets at the OOD detection for generative models, and we incorporate a similar idea of "regret" between the likelihood under the trained model and the optimal likelihood for the individual test point obtained by optimizing on the single test input. I think the connection between these two papers is interesting, and they combinedly show that the idea of optimizing test data is a good method to estimate uncertainty.

---

### Author Response · Authors · 2020-11-19
**Updated paper with Bibas et al (2019) comparisons**

As suggested by the reviewers, we have added comparisons on the MNIST domains with the much more computationally expensive method of Bibas et al, which directly uses the training set in finetuning the parameters for each query point/label instead of an approximate posterior as in ACNML. We have updated Table 1 in the experiments section (with the method titled naive CNML), the timing results in Table 2, and include more detailed comparisons between the predictions of naive CNML and ACNML in Appendix C.

We first note that this naive CNML instantiation with a small feedforward network on MNIST, which finetuned with 5 epochs over the training set, took **70 seconds** per datapoint, compared to **0.08 seconds** for ACNML, taking **almost 1000 times** as long for inference. Overall, we find that, at least with the diagonal Gaussian posterior approximation obtained via Bayes-by-Backprop, ACNML and the naive CNML implementation perform somewhat differently. Naive CNML achieves the best NLL and accuracy among all methods on the in-distribution test set. On the OOD rotated images, naive CNML does output more conservative predictions than the MAP, resulting in improved NLL and calibration over the MAP as we would expect. ACNML however has overall more conservative predictions than naive CNML, which results in worse performance in-distribution, but better NLL and calibration on the OOD images.

We again thank all the reviewers for their helpful feedback, and hope the updates we have made incorporating that feedback have improved the paper. We ask all reviewers to please let us know if their concerns have been addressed, and if not, which concerns or questions remain.

---

### Decision · Program_Chairs · 2021-01-07
**Final Decision**

**Decision:**

Reject

**Comment:**

The paper had four borderline reviews with none enthusiastic about championing the merits of the paper. While it was felt that the extension of an existing technique to deep learning via amortization is a useful procedure, it is also not very novel and the experiments didn't demonstrate a significant leap in performance.